# Differentiation of Human-Induced Pluripotent Stem Cell-Derived Endocrine Progenitors to Islet-like Cells Using a Dialysis Suspension Culture System

**DOI:** 10.3390/cells10082017

**Published:** 2021-08-07

**Authors:** Hyunjin Choi, Marie Shinohara, Masato Ibuki, Masaki Nishikawa, Yasuyuki Sakai

**Affiliations:** 1Department of Bioengineering, Graduate School of Engineering, The University of Tokyo, Tokyo 113-8654, Japan; 2Institute of Industrial Science, The University of Tokyo, Tokyo 153-8505, Japan; marie-s@iis.u-tokyo.ac.jp (M.S.); ma.ibu.jpn@gmail.com (M.I.); 3Kaneka Corporation, Osaka 530-0005, Japan; 4Department of Chemical System Engineering, Graduate School of Engineering, The University of Tokyo, Tokyo 113-8654, Japan; masaki@chemsys.t.u-tokyo.ac.jp (M.N.); sakaiyasu@chemsys.t.u-tokyo.ac.jp (Y.S.)

**Keywords:** pancreatic β cell, islet-like cell, hiPSC, differentiation, dialysis, suspension culture, cost reduction

## Abstract

The production of functional islet-like cells from human-induced pluripotent stem cells (hiPSCs) is a promising strategy for the therapeutic use and disease modeling for type 1 diabetes. However, the production cost of islet-like cells is extremely high due to the use of expensive growth factors for differentiation. In a conventional culture method, growth factors and beneficial autocrine factors remaining in the culture medium are removed along with toxic metabolites during the medium change, and it limits the efficient utilization of those factors. In this study, we demonstrated that the dialysis suspension culture system is possible to reduce the usage of growth factors to one-third in the differentiation of hiPSC-derived endocrine progenitor cells to islet-like cells by reducing the medium change frequency with the refinement of the culture medium. Furthermore, the expression levels of hormone-secretion-related genes and the efficiency of differentiation were improved with the dialysis suspension culture system, possibly due to the retaining of autocrine factors. In addition, we confirmed several improvements required for the further study of the dialysis culture system. These findings showed the promising possibility of the dialysis suspension culture system for the low-cost production of islet-like cells.

## 1. Introduction

Production of insulin-secreting pancreatic islet-like cells by inducing the differentiation of human-induced pluripotent stem cells (hiPSC) has attracted attention as alternative cell sources for the therapeutic use for type 1 diabetes and disease modeling [1,2]. Through considerable efforts and studies, several approaches for the induction of differentiation have been developed. One of the most successful strategies is several days of cultivation mimicking the environment of pancreas development with a medium containing several kinds of growth factors and chemical compounds [3,4], in the order of definitive endoderm (DE), primitive gut tube (PGT), posterior foregut tube (PFG), pancreatic progenitor cells (PP), endocrine progenitor (EP), and pancreatic islet-like cells consisting of α cells, δ cells, and β cells [5,6]. However, this strategy has been limited by the high production cost because the price of growth factors is expensive and a considerable amount of cells are required for the applications [4,7]. In order to reduce expensive growth factors, various studies using inexpensive small molecules or gene transfer have been tried, but growth factors have not yet been completely replaced [8,9,10] Therefore, the development of a large-scale culture system that enables the reduction of growth factors is considered to be a promising strategy for reducing the production cost of islet-like cells.

For a large-scale culture, aggregate suspension culture is thought to be suitable because of the potential of increasing cell density and culture scale. Furthermore, it is reported that insulin secretion capability of differentiated pancreatic β cells is improved in aggregate suspension culture compared with an adherent culture [11,12,13]. However, as more cells are cultured in a suspension culture, culture environments can be easily deteriorated by nutritional depletion and the accumulation of toxic metabolites. Particularly, the accumulation of lactate in the culture medium negatively affects cells, regardless of cell types, by decreasing pH [14,15,16], thus the removal of lactate during the culture is essential. Moreover, the used culture medium after cultivation contains remaining growth factors and various autocrine factors secreted by cells. Those factors are difficult to utilize due to the periodic medium replacement to remove lactate.

In our previous report, we designed a simple dialysis suspension culture system and demonstrated its effective usage in the differentiation of hiPSCs to DE cells [17]. The dialysis culture was possible to refine the culture medium by removing lactate while retaining macromolecules such as growth factors and autocrine factors in the culture medium, which enabled the reduction of the medium replacement frequency and minimized the usage of growth factors. However, this system has not been applied to the other differentiation stages of pancreatic islet-like cells. Among those stages, the differentiation of EP cells into pancreatic islet-like cells is especially important, because this stage requires the second-highest cost in the whole stages next to DE differentiation, and the maturity of pancreatic β cells in this stage affects the capability of the insulin-secretion of pancreatic β cells [4,11,12].

In this study, we investigated the feasibility of the dialysis suspension culture system in the differentiation of EP cells to pancreatic islet-like cells.

## 2. Materials and Methods

### 2.1. Preparation of Dialysis Suspension Culture Plate

The dialysis culture plate was prepared in the same way as the previous study [17]. Briefly, the original membrane of a 6-well culture insert (Corning, New York, NY, USA) was replaced with a regenerated cellulose dialysis membrane (Spectrum Labs, San Francisco, CA, USA), which had a 3 kDa molecular weight cut off (MCWO). The dialysis membrane combined insert cup was inserted in the deep well plate (Corning,) (Figure 1a). During the differentiation, 2.5 mL of differentiation medium was added to the culture insert and 14.5 mL of dialysis medium was added in the deep well (Figure 1b). The molecule larger than 3 kDa remained in the culture insert during the culture on an orbital lab shaker. To compare the dialysis culture with a conventional suspension culture without dialysis, the dialysis membrane of the culture insert was sealed with polyethylene film (Diversified Biotech, Dedham, MA, USA) to block the mass transfer by the dialysis.

### 2.2. Culture of hiPSC and hiPSC Aggregates Formation

The human iPS cell line TkDN4-M was provided by the Stem Cell Bank at the Institute of Medical Science, The University of Tokyo. Cells were cultured on a 6-well tissue culture treated plate (Wako) and coated with vitronectin fragment (VTN-N; Gibco, New York, NY, USA) in Essential 8 (E8; Gibco)-supplemented 1% penicillin streptomycin amphotericin B suspension (PSA; Wako, Osaka, Japan). For the subculture, cultured hiPSCs were dissociated by Accutase (Innovative Cell Technologies, San Diego, CA, USA), and single cells were seeded at 1.0 × 10^4^ cells/cm^2^ in E8 medium containing 1% PSA, 10 mM Rho-associated kinase inhibitor (Y-27632; Wako). Cells were incubated at 37 °C in a humidified atmosphere with 5% CO_2_, and the culture medium was changed daily with E8 medium containing 1% PSA.

For the cell aggregate formation, dissociated cells were seeded at 7.5 × 10^5^ cells/mL with E8 medium containing 1% PSA, 5 mg/mL bovine serum albumin (BSA; Proliant Biologicals, Ankeny, IA, USA), and 10 mM Y-27632 in an Erlenmeyer flask (Corning). The Erlenmeyer flask was rotated on an orbital shaker at 90 rpm. Cells were incubated at 37 °C in a humidified atmosphere with 5% CO_2_, and the culture medium was changed daily with E8 medium containing 1% PSA and 5 mg/mL BSA. After three to five days of culture, the aggregates were formed.

### 2.3. Preparation of Endocrine Progenitor Cells

EP cells for the investigation of the dialysis suspension culture were prepared by hiPSCs differentiation. For the induction of hiPSCs differentiation into EP cells, hiPSC aggregates were precultured with DMEM/Ham’s F-12 medium (Wako) containing 1% PSA, 5 μM Y27632, 20% knockout serum replacement (KSR; Gibco), 1% non-essential amino acids (NEAA; Wako), and 55 μM 2-mercaptoethanol (2-ME; Gibco) for one day. The differentiation of EP was performed as described previously with some modification in 5 stages [4,12]:

At stage 1, for the differentiation of hiPSCs into definitive endoderm (DE), hiPSCs aggregates were cultured for four days with daily medium changes in RPMI 1640 (Wako) containing 5 mg/mL BSA, 1 mM sodium pyruvate, 1% NEAA, 0.4% PSA, 80 ng/mL recombinant human activin A (R&D Systems, Minneapolis, MN, USA), 55 μM 2-ME, 50 ng/mL recombinant human fibroblast growth factor 2 (FGF2; Reprocell, Kanagawa, Japan), and 20 ng/mL recombinant human bone morphogenetic protein 4 (BMP4; Miltenyi Biotec, Teterow, Germany). 3 μM CHIR99021 (Wako) was added on day 1 whereas 1% KSR was added on day 4.

At stage 2, for the differentiation of the primitive gut tube (PGT), DE cell aggregates were cultured for two days in RPMI 1640 containing 5 mg/mL BSA, 1 mM sodium pyruvate, 1% NEAA, 0.4% PSA, 50 ng/mL FGF2, 50 ng/mL recombinant human FGF7 (Miltenyi Biotec), and 2% B-27 supplement (Gibco);

At stage 3, for the differentiation of the posterior foregut (PFG), PGT cell aggregates were cultured for four days, and the medium was replaced at day 2. The cell aggregates were cultured in DMEM-high glucose (Wako) supplemented with 1% NEAA, 0.4% PSA, 2% B-27, 0.67 μM EC23 (Santa Cruz Biotechnology, Dallas, TX, USA), 1 μM Dorsomorphin (Wako), 10 μM SB431542 (Wako), and 0.25 μM SANT1 (Wako);

At stage 4, for the differentiation of the pancreatic progenitor (PP), PFG cell aggregates were cultured for three days, and the medium was replaced at day 2. The cell aggregates were cultured in DMEM-high glucose supplemented with 1% NEAA, 0.4% PSA, 2% B-27, 50 ng/mL recombinant human FGF10 (Wako), 0.5 μM EC23, 1 μM Dorsomorphin, 5 μM Alk5 inhibitor II (Rep Sox; Sigma Aldrich, St. Louis, MO, USA), 0.25 μM SNAT1, and 300 nM indolactam V (ILV; Cayman Chemical, Ann Arbor, MI, USA).

At stage 5 for the differentiation of the endocrine progenitor (EP), PP cell aggregates were cultured for 3 days, and the medium was replaced at day 2. The cell aggregates were cultured in Advanced DMEM (Gibco) supplemented with 2 mM L-glutamine (Wako), 0.4% PSA, 2% B-27, 50 ng/mL exendin 4 (EX4; Wako), 0.2 μM EC23, and 1 uM RepSox, 0.25 μM SANT1.

### 2.4. Differentiation of Islet-like Cells with Dialysis Suspension Culture System

EP cell aggregates were collected and moved to the dialysis suspension culture plate in 2.4 × 10^5^ cells/mL on the final day of stage 5 (Figure 1b). During the differentiation of EP cells to islet-like cells in stage 6, aggregates were cultured in the culture insert with a 2.5 mL differentiation medium for six days. The differentiation medium was Advanced DMEM containing macromolecules that have larger molecular weight than MWCO of the dialysis membrane (3 kDa); 10 ng/mL BMP4 (34 kDa), 10 ng/mL FGF2 (17 kDa), 50 ng/mL recombinant human hepatocyte growth factor (HGF; Miltenyi Biotec) (70 kDa), 50 ng/mL insulin-like growth factor 1 (IGF1; Peprotech, East Windsor, NJ, USA) (7.7 kDa), 50 ng/mL EX4 (4 kDa), supplement; 2 mM L-glutamine, 0.4% PSA, 2% B-27 and small molecules; 5 μM RepSox, 10 μM Forskolin (FUJIFILM Wako Pure Chemical Corporation, Mie Pref, Japan), and 5 mM Nicotinamide (Sigma Aldrich). Under the culture insert, a 14.5 mL dialysis medium filled the deep well. The dialysis medium was Advanced DMEM containing supplement; 2 mM L-glutamine, 0.4% PSA, 2% B-27 and small molecules; 5 μM RepSox, 10 μM Forskolin, and 5 mM Nicotinamide, without growth factors.

To evaluate the dialysis effect on the differentiation in stage 6, EP aggregates were cultured in the four different culture conditions named Control, DC(2), DC(1), and DC(0) (Figure 1d). In the Control group, the dialysis culture insert was sealed so that the medium was not mixed between the differentiation medium and the dialysis medium. The differentiation medium was refreshed every other day. The dialysis culture medium was not changed. In the DC(2), DC(1), and DC(0) groups, the differentiation medium, and the dialysis medium were able to interact through the dialysis membrane. The differentiation medium and the dialysis medium were refreshed twice, once, and none in DC(2), DC(1), and DC(0), respectively.

### 2.5. Measurement of Glucose and Lactate

To evaluate the refinement effect of dialysis operations during the culture, differentiation medium and dialysis medium were collected at every medium change. The glucose and lactate levels of each medium were measured by a bioanalyzer (Oji scientific instruments, Amagasaki, Japan).

### 2.6. RT-qPCR Analysis

On day 0 and day 6 of islet-like cell differentiation, cell aggregates were collected with TRIZol reagent (Thermo Fisher Scientific, Waltham, MA, USA). Total RNA was isolated by adding chloroform (Wako) with centrifugation of 15,000× *g* for 15 min. The 2-propanol (Wako) was added to collected supernatants and centrifuged at 15,000× *g* for 15 min to make a purified RNA pellet. The RNA pellet was washed with 75% ethanol and dissolved in RNase-free water. The concentration of RNA was measured by NanoDrop (Shimadzu, Kyoto, Japan) and 100 ng of RNA from each sample was reverse–transcribed by Prime ScriptTM Reverse Transcriptase (Takara Bio Inc., Kusatsu, Japan). After reverse transcription, transcribed complementary DNA samples were quantitated by SYBR Green gene expression assays and detected by StepOne Plus (Thermo Fisher Scientific). Target genes primer sequences are shown in Appendix A.

### 2.7. Immunostaining

Differentiated islet-like cells were collected and fixed with 4% paraformaldehyde. Fixed cell aggregates were washed with phosphate buffer saline solution (PBS) 3 times for 15 min and permeabilized with 1% Triton X100 in PBS for 1 h at room temperature. Permeabilized cell aggregates were washed 3 times with PBS for 15 min and blocked with 6% BSA in PBS overnight at 4 °C. Immunostaining was performed using the following primary antibodies, mouse anti-insulin (1:500; Abcam, Cambridge, UK), and mouse anti-glucagon (1:500; Abcam) diluted in PBS containing 3% BSA (PBSB) and incubated for 48 h at 4 °C. After washing 3 times for 30 min with PBS containing 0.2% Tween 20 (PBST), cells were incubated with secondary antibodies (diluted with PBSB), Alexa Fluor 647-conjugated donkey anti-mouse IgG (1:500; Abcam) overnight at 4 °C in the dark. After the immunostaining, the nuclei were stained with 4′-6′-diamidino-2-phenylindole (DAPI; 1:1000; Dojindo, Kumamoto, Japan). Then, cell aggregates were cleared by tissue clearing reagent (Visikol, Hampton, NJ, USA) to observe the cross-section of dense cell aggregates. The fluorescent images were obtained with a confocal laser-scanning microscope (IX-81, Olympus, Tokyo, Japan) and the percentage of positive cell area to the cross-sectional area of islet-like cells was calculated in each image using Image J software (National Institute of Mental Health, Bethesda, MD, USA).

### 2.8. Glucose-Stimulated C-Peptide Secretion Assays

After the differentiation, islet-like cells in each culture condition were collected and glucose-stimulated. C-peptide secretion assay was performed to determine glucose-stimulated insulin secretion (GSIS). Glucose stimulation was performed in two phases and low high glucose stimulation by repeating the following operations. Cell aggregates were washed twice with DMEM (no glucose) containing 10 mM HEPES, 0.1% BSA, and preincubated at 37 °C for 30 min in DMEM (2 mM glucose) containing 10 mM HEPES, 0.1% BSA. Cell aggregates were washed twice with DMEM (no glucose) containing 10 mM HEPES, 0.1% BSA, and then incubated at 37 °C for 1 h in 1 mL DMEM (2 mM glucose) containing 10 mM HEPES, 0.1% BSA. The supernatant of the medium was collected after gentle mixing to make the C-peptide concentration homogeneous. Cell aggregates were washed twice with DMEM (no glucose) containing 10 mM HEPES, 0.1% BSA, and then incubated at 37 °C for 1 h in 1 mL DMEM (20 mM glucose) containing 10 mM HEPES, 0.1% BSA, and the supernatant was collected. The C-peptide concentration of the sampled supernatant in each phase was measured using the human C-peptide ELISA kit (Mercodia, Winston Salem, NC, USA) following the manufacturer’s protocol.

### 2.9. Statistical Analysis

The data results of this study were obtained from two independent experiments and expressed as the mean ± standard deviation of the mean (SD). Statical analyses were performed using the Student’s *t*-test to evaluate differences. One-way ANOVA, with the post hoc Tukey HSD test, were performed for the comparison between more than two groups. A value of *p* < 0.05 was considered significant.

## 3. Results

### 3.1. Dialysis Suspension Culture Refined the Medium by Supplied Nutrient and Removed Waste

The glucose and lactate levels of the medium were measured to confirm the refinement effect of the dialysis culture system. In the dialyzed culture conditions such as DC(2), DC(1), and even in DC(0) without a medium change, showed a conducive culture environment with higher glucose and lower lactate levels than the Control culture without dialysis operations (Figure 2a,b). Furthermore, accumulation in the dialysis medium were observed in dialyzed culture conditions, indicating continuous lactate removal occurred by dialysis operation (Figure 2d). In addition, the glucose concentration of the differentiation medium and the dialysis medium in dialyzed culture conditions closed to fresh medium, indicated that glucose supply was abundant in the dialysis culture system (Figure 2a,c).

### 3.2. Dialysis Suspension Culture Did Not Affect the Number of Differentiated Islet-like Cells

Prepared hiPSCs-derived EP cell aggregates were moved to the dialysis suspension culture system, and differentiation of islet-like cells was performed with 2.4 × 10^5^ cells/mL cell density in the culture insert (Figure 1b). After stage 6, the differentiated cell aggregates were collected and dissociated for cell count. There were no significant differences shown in the number of produced cells between Control, DC(2), DC(1), and DC(0) (Figure 3).

### 3.3. Dialysis Suspension Culture System Improves the Quality of Islet-like Cells

To evaluate the gene expression of differentiated islet-like cells cultured in each culture condition, the mRNA levels were measured by RT-qPCR (Figure 4). The mRNA expression of *NGN3* was decreased in all conditions on day 6 compared to EP cells. *NGN3* starts to express from the early endocrine progenitor and decreases with the maturation of islet-like cells [18]. In contrast, the expression levels of pancreatic genes such as *PDX1*, *NKX6.1*, *MAFA*, *INS*, *GCG*, *SST*, *GLUT2*, and *PCSK1* were upregulated on day 6 of stage 6 in all conditions compared to EP cells. These results suggest that the differentiation of islet-like cells was successful. DC(1) and DC(0), with less medium change frequency, expressed significantly higher levels of pancreatic endocrine hormones *INS* (insulin) and *GCG* (glucagon) than Control and DC(2), with conventional medium change frequency. Furthermore, expression levels of glucose sensing and metabolism-related genes such as *GLUT* (glucose transporter) *1*, *GLUT2*, and *GCK* (glucokinase) in DC(0) tended to be higher than in the Control. *GLUT2,* particularly, was expressed significantly higher in DC(1) and DC(0) than in the Control, whereas there were no significant differences showed in *PDX1*, *NKX6.1*, *MAFA*, and *PCSK1*.

The characteristic of islet-like cells was examined by immunostaining in each culture condition. We observed insulin and glucagon positive cells of islet-like cells in all culture conditions, and we found the positive cell area of DC(1) and DC(0) was increased (Figure 5). From the measurement of insulin and glucagon positive cell area, the percentage of insulin-produced cell area of DC(1) (20.6 ± 5.8%) and DC(0) (24.1 ± 7.9%) were significantly increased compared to the Control (8.6 ± 4.4%) (Appendix A). In addition, the glucagon-produced cell area of DC(0) (1.78 ± 1.02%) showed a significantly larger area than the Control (0.52 ± 0.36%) (Appendix A). These results indicate that more cells in DC(0) produced insulin or glucagon, following the results of mRNA expression levels of *INS* and *GCG*.

To confirm the function of pancreatic β cells in the differentiated islet-like cells, we performed a low high glucose-stimulated C-peptide secretion assay to evaluate GSIS (Figure 6). In the first glucose stimulation, we could not observe a significant increase of C-peptide secretion by high glucose in all culture conditions (Appendix A). In our experience, sometimes differentiated islet-like cells showed unstable C-peptide secretion in the first glucose stimulation, thought to be due to their immatureness. In the second glucose stimulation of GSIS, we observed the fold increase of C-peptide secretion response to high glucose and only DC(0) showed a significant increase of C-peptide secretion (Figure 6). In addition, DC(1) and DC(0) showed the tendency of higher secretion of C-peptide in each low and high glucose concentration than DC(2) and Control. Then, we calculated the ratio of C-peptide secretion in high glucose to low glucose, and the results were Control: 1.26 ± 0.76, DC(2): 1.35 ± 0.64, DC(1): 1.18 ± 0.35, DC(0): 1.57 ± 0.57 (Appendix A).

## 4. Discussion

The generation of islet-like cells from hiPSCs is a promising cell source but one of the drawbacks is the expensive production cost due to the extensive usage of growth factors. According to our cost calculation based on commercially available reagents, approximately 71% of the total cost of the differentiation medium is required for the growth factors (Appendix A). In addition, in the conventional culture method, growth factors could not be fully utilized due to frequent medium changes required for lactate removal.

From our previous report, we designed a simple dialysis suspension culture system and demonstrated the possibility of reducing the usage of growth factors by reducing medium change frequency in the differentiation of hiPSCs to DE cells [17]. In this study, continuing from the previous report, we investigated the performance of dialysis suspension culture systems in the differentiation of hiPSC-derived EP cells to pancreatic islet-like cells. Our findings suggest that this system can reduce the medium change frequency by maintaining culture environments with continuous glucose supply and lactate removal. Additionally, the differentiation efficiency of islet-like cells was improved, possibly due to the retaining and full utilization of external and autocrine growth factors in the culture medium during the differentiation.

Several studies reported that the decrease in pH from excess lactate accumulation has negative effects on the cells regardless of cell types, hence, preventing the accumulation of lactate is important in cell quality control. The limit of lactate concentration is 15 mM for human ES cells and hiPSC [14,16], and 22 mM for other differentiated somatic cells such as Chinese hamster ovary cells [15,19]. From the experiments, we observed continuous lactate removal during the differentiation with the dialysis suspension culture system (Figure 4). Especially, we successfully reduced the medium change frequency from twice to once and twice to zero in DC(1) and DC(0), respectively, which allowed the reduction of growth factors usage. We used only half of the growth factors for DC(1) and one-third of the growth factors for DC(0) compared to the control culture. Moreover, considering that the lactate concentration of the medium in DC(0) was sufficiently low without medium change, the dialysis suspension culture still had the capacity to culture more cells at a higher density, as was also shown in other reports. In this study, we generated approximately 2 × 10^5^ cells/mL of islet-like cells from 2.4 × 10^5^ cells/mL of EP cells. However, as the yields of the suspension culture system are reported to be 1.7 × 10^6^ cells/mL and 1.6 × 10^6^ cells/mL by Yabe et al. and Mihara et al., respectively [4,20], the cell density of our study was relatively low, thus, further investigations of dialysis suspension culture systems in higher cell density are necessary to confirm the feasibility of dialysis suspension culture systems in the differentiation of islet-like cells.

From the experiments, we demonstrated that the dialysis suspension culture system improved the differentiation efficiency of islet-like cells while reducing the usage of growth factors. The expression levels of hormone-secretion-related genes such as *INS* and *GCG* were upregulated in DC(1) and DC(0). Consistent with these results, the ratio of insulin or glucagon positive cells, and the secretion amount of insulin of islet-like cells in DC(1) and DC(0) were increased (Figure 4, Figure 5 and Figure 6). The *INS* gene encodes insulin secreted by β cells, which is the main target in this research. The *GCG* gene encodes glucagon secreted by α cells, which is also important for glycemic control, as the glucagon-secreting α cells can improve the function of β cells [21,22]. In addition, the GSIS of islet-like cells in DC(0) showed an improved response to the change of glucose concentration. The fold increase of insulin secretion-response to high glucose concentration in DC(0) was the highest (Appendix A), and only DC(0) showed a significant increase (Figure 6). This improvement is possibly due to the upregulations in glucose-sensing and glucose metabolism-related genes, such as *GLUT1*, *GLUT2*, and *GCK* (Figure 4). Because the GSIS is regulated by the glucose metabolism of pancreatic β cells [23,24,25], the glucose sensing and transport by the glucose transporter (encoded by *GLUT*) [23,25], and phosphorylation of glucose for the glycolysis through the glucokinase (encoded by *GCK*), is important for the GSIS [26,27]. Furthermore, there were no significant changes in the expression levels of *PDX1 NKX6.1*, *MAFA*, involved in β cell development, and *PCSK1* involved in the insulin process, which indicates that the improvement of the GSIS in DC(0) is mainly due to the maturation of the glucose-sensing and glucose metabolism of β cells. However, although the expression of *GLUT2* was increased in DC(1), the fold increase of the insulin secretion response to high glucose was not significant. One possible reason is that the islet-like cells in DC(1) secreted excess insulin at the first glucose stimulation, which depleted insulin to affect the insulin secretion at the second glucose stimulation. The total C-peptide secretion amount at the first glucose stimulation was 1.18 ± 0.17-fold in Control, 1.40 ± 0.31-fold in DC(2), 1.78 ± 1.29-fold in DC(1), and 1.14 ± 0.45-fold higher in DC(0), compared to the second stimulation. A possible reason for this result is that in DC(1), the increase of *GLUT2* led to the maturation of the glucose transporter to induce active glucose transport, but the immaturity of the glucose metabolism due to the insufficient expression of *GCK* caused excessive insulin secretion. However, we found that the maturity of islet-like cells in the Control and DC(2) culture was low. In the previous papers that we referred to by the differentiation protocol of this study, they obtained approximately 30% of C-peptide and 5% of glucagon positive cells from the differentiated islet-like cells, and only the DC(0) culture showed a similar result in this work, which is possibly due to the differences in the cell line or technique of this study. It is expected that extending the culture days of differentiation can improve the maturity of islet-like cells in all culture conditions. However, since the long-term dialysis culture may lead to the depletion of growth factors and the accumulation of cytotoxic macromolecules such as proinflammatory cytokines that cannot be removed by dialysis, further studies on the appropriate medium change frequency in the dialysis culture system is necessary for the long-term culture.

A conceivable reason for those improvements is the accumulation of autocrine factors during the differentiation, and one possible autocrine factor is glucagon-like peptide-1(GLP-1). GLP-1 is secreted from early α cells and is known to play an important role in the development of β cells and insulin secretion [28,29,30,31]. Because the molecular weight of GLP-1 (3.2 kDa) is larger than the MCWO of the dialysis membrane (3 kDa), GLP-1 is expected to accumulate in the culture medium during the differentiation.

The dialysis suspension culture system showed different trends of effects on the differentiation of islet-like cells from DE differentiation [17]. Under the DC(2) condition, the yield of the product cells was improved only in the DE differentiation, but not the differentiation of islet-like cells, while the improvements of gene expression patterns and functions under the DC(0) condition were only observed in the differentiation of islet-like cells. These findings suggest that the optimum experimental technique for the dialysis suspension culture system differs depending on the characteristics of cells in each differentiation stage.

In this study, we demonstrated that the dialysis suspension culture could generate more functional islet-like cells with one-third of the growth factors. However, the use of the B-27 supplements in the differentiation protocol limited the effect of the cost reduction of our system. B-27 is a defined supplement that includes various small molecules and proteins and is widely used in the differentiation protocols of iPSCs to improve cell survival [32,33]. Because B-27 includes small molecules under the size of the MCWO of the membrane in our system, B-27 was required for both the differentiation medium and the dialysis medium. Based on the composition of B-27 and our cost calculations, proteins such as superoxide dismutase (16 kDa) and bovine serum albumin (65 kDa), accounted for the majority of the cost [34,35], suggesting the possibility of further cost reduction by separating B-27 into proteins and small molecules.

In this paper, we confirmed several advantages of the dialysis culture system in the differentiation of hiPSC-derived EP cells into islet-like cells but also identified several improvements required for further investigations in the dialysis culture system. According to the ratio of insulin-positive cells and GSIS in Control, DC(2), and DC(1) culture conditions, the maturity of islet-like cells in those culture conditions were low, thus, it seems necessary to conduct islet-like cell differentiation under higher maturity, for a deeper understanding of cell function in the biological improvement of the dialysis culture system. However, even though such weaknesses existed in this paper, this paper still shows promising engineering advantages of the dialysis culture system as a cultivation technique that can reduce the production cost of hiPSC-derived islet-like cells by fully utilizing growth factors and autocrine factors. Particularly in the DC(0) culture condition, we reduced the use of growth factors by one-third, without the functional decline of islet-like cells, and increased differentiation efficiency. Therefore, we believe that this paper shows the direction and possibility for further investigations of the dialysis culture system in islet-like cell production.

## 5. Conclusions

In this work, we demonstrated the dialysis suspension culture system in the differentiation of hiPSC-derived EP cells to islet-like cells, successfully reduced the usage of growth factors to one-third, and with higher differentiation efficiency. The dialysis culture refined the culture medium and resulted in the reduction of medium change frequency without loss of cell yield. Furthermore, in the dialysis suspension culture system, we observed the upregulations in the expression levels of hormone-secretion-related genes, and insulin secretion functions, due to the increase of differentiation efficiency possibly due to the retaining and utilizing of growth factors and autocrine factors. In addition, we found several improvements requiring further study. Therefore, together with further modifications in the dialysis culture system, we believe that this study will improve the manufacturability of pancreatic islet-like cells.

## Figures and Tables

**Figure 1 cells-10-02017-f001:**
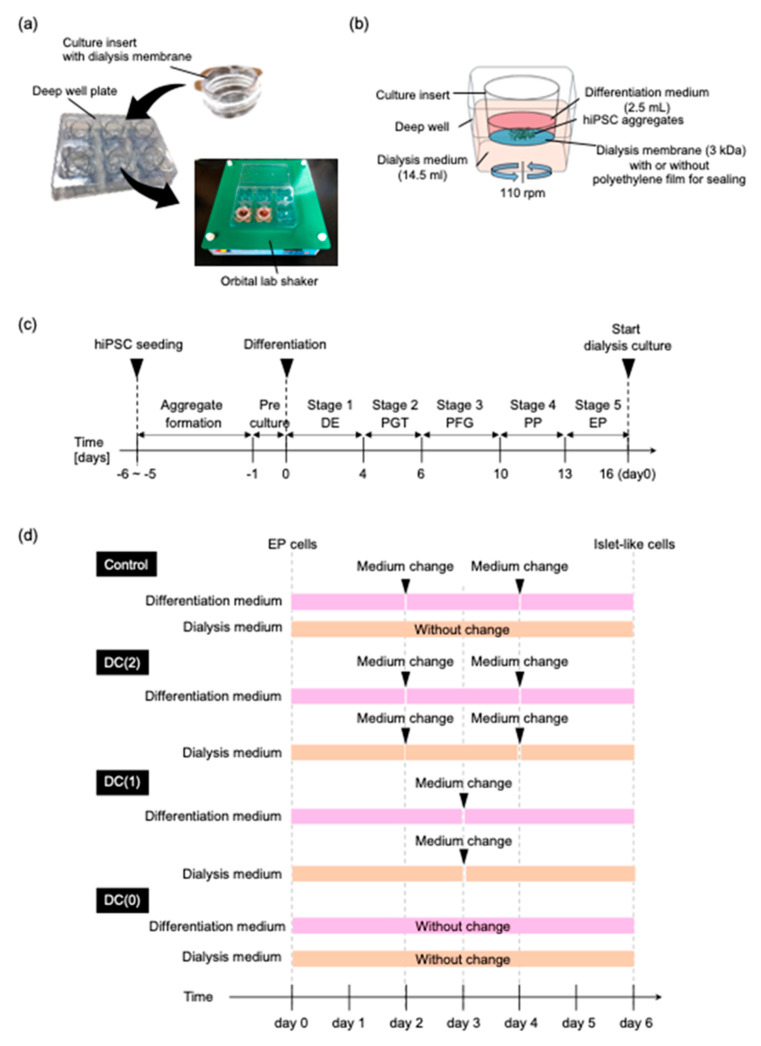
Dialysis suspension culture plate was used for the differentiation of EP cells to islet-like cells. (**a**) Photograph of the dialysis suspension culture plate. (**b**) Schematic diagram of dialysis suspension culture plate for the differentiation of endocrine progenitor cells into islet-like cells. (**c**) Overview of the differentiation schedule for preparation of EP cells. (**d**) Schedule of differentiation of EP cells into islet-like cells, and medium change interval of each culture’s conditions in Stage 6.

**Figure 2 cells-10-02017-f002:**
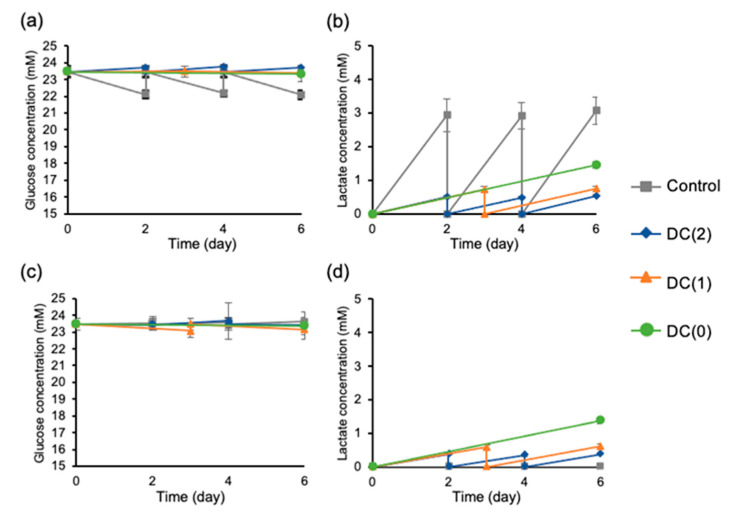
Glucose and lactate concentration of differentiation medium and dialysis medium were measured at every medium change during the differentiation. (**a**) Glucose concentration of differentiation medium, (**b**) lactate concentration of differentiation medium, (**c**) glucose concentration of dialysis medium, and (**d**) lactate concentration of dialysis medium. Control: without dialysis, medium change 2 times, DC(2): with dialysis, medium change 2 times, DC(1): with dialysis, medium change 1 time, DC(0): with dialysis, without medium change. Data represent mean ± SD of *n* = 5–6 from two independent experiments.

**Figure 3 cells-10-02017-f003:**
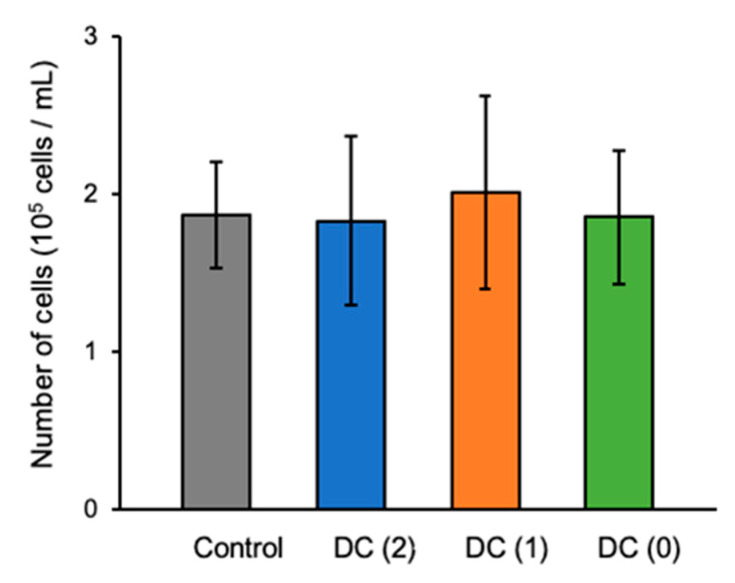
The final cell density of differentiated islet-like cells in each culture’s conditions. Data represent mean ± SD of *n* = 5–6 from two independent experiments.

**Figure 4 cells-10-02017-f004:**
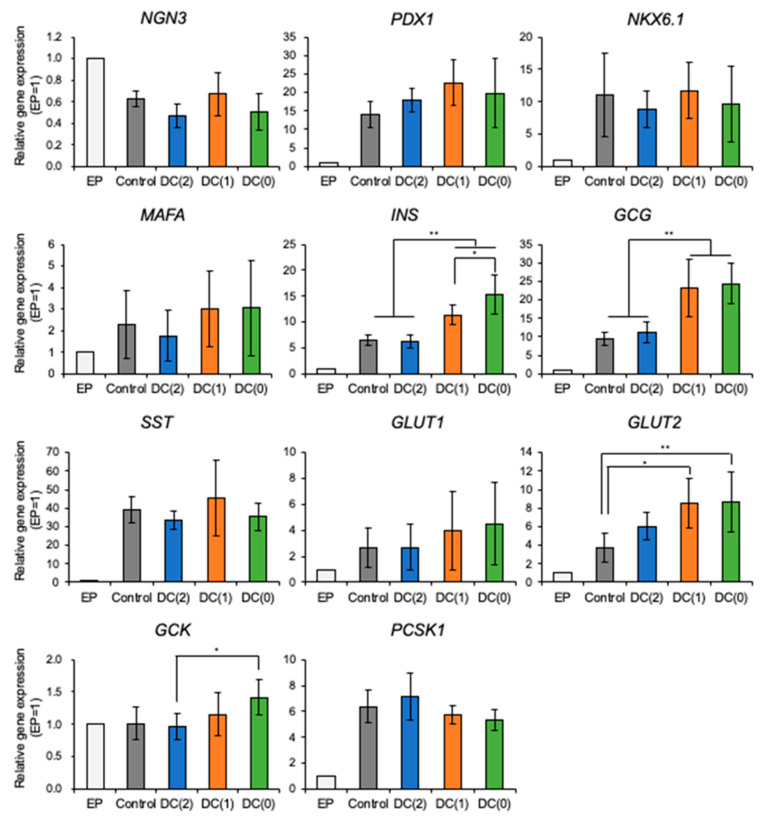
Relative gene expression levels of *NGN3*, *PDX1*, *NKX6.1*, *MAFA*, *INS*, *GCG*, *SST*, *GLUT1*, *GLUT2*, *GCK*, and *PCSK1* in differentiated EP cells and islet-like cells were measured by RT-qPCR. Data represent mean ± SD of *n* = 5–6 from two independent experiments. Statistical analysis as determined by one-way ANOVA with post hoc Tukey HSD test. (* *p* < 0.05, ** *p* < 0.01).

**Figure 5 cells-10-02017-f005:**
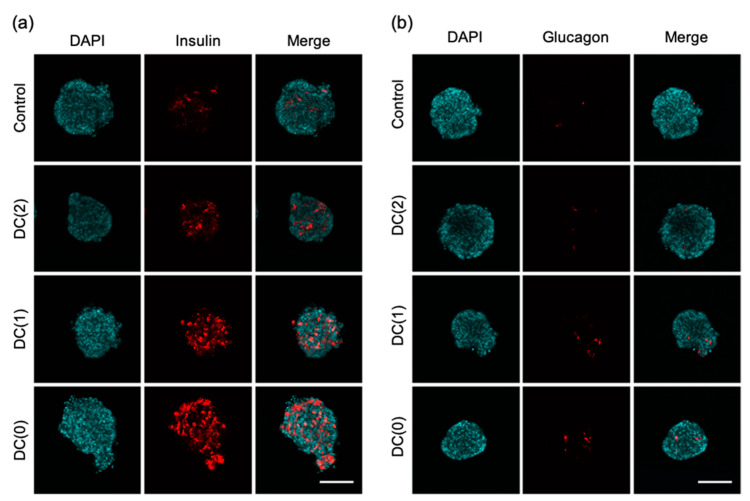
Immunostaining of islet-like cells after the differentiation in each culture condition. (**a**) DAPI, insulin, merge, (**b**) DAPI, glucagon, merge. (Scale bar = 100 μm).

**Figure 6 cells-10-02017-f006:**
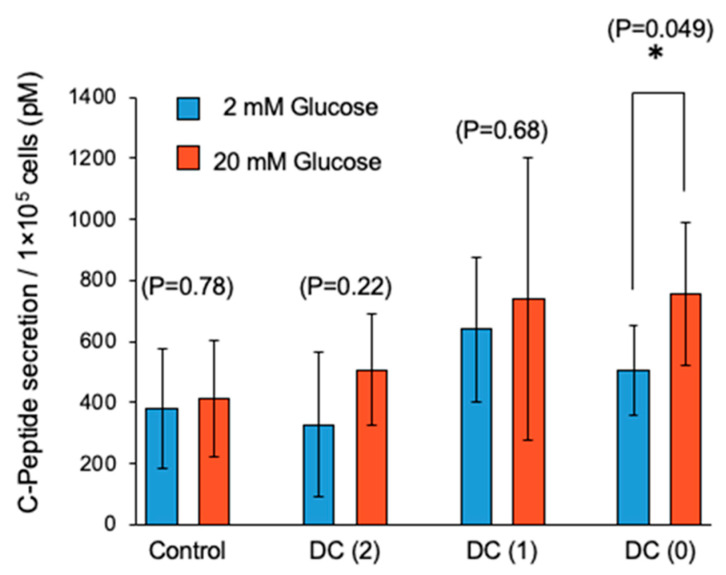
Glucose-stimulated C-peptide secretion of differentiated islet-like cells in second phase glucose stimulation. Data represent mean ± SD of *n* = 5–6 from two independent experiments. Statistical analysis as determined by Student’s *t*-test. (* *p* < 0.05).

## Data Availability

The data presented in this study are available on request from the corresponding author.

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
