# Peer review of "Differentiation of Human-Induced Pluripotent Stem Cell-Derived Endocrine Progenitors to Islet-like Cells Using a Dialysis Suspension Culture System"

_cells, 2021, doi:10.3390/cells10082017_

Round 1
Reviewer 1 Report
Choi et al generated a simple dialysis culture system for islet-like cells differentiation from iPSCs and significantly reduced the cost of islet-like cell differentiation. This low-cost system may have potential application in islet-like cell differentiation without affecting differentiation efficiency and function. There are some suggestions:
- Did the author test the markers for islet-like cells in the culture by immunostaining to visualize the islet-like cells? It would be better to visualize the islet-like cells in the groups to see whether the different conditions affect the differentiation of stem cells to islet-like cells.
- For the mRNA expression, what time point the authors collected samples? In Method 2.6, the author described “Cells were collected on day 6 in the stage 6 from each group after differentiation with TRIZol reagent”. This is confusing, it looks at day 6 of the stage 6? Or the author could use total days from starting.
- Please include the time point for C-peptide.
- The authors evaluated the short-term, how about the effect of long-term using this system?
Author Response
Answers to reviewer #1
Dear reviewer,
Thank you very much for appreciating our work and your valuable comments and suggestions. We have attempted to answer your comments into consideration. Below we have replied to each comment in turn and marked the modified part. We look forward to your reply.
Best regards,
Hyunjin Choi
Point-to-point response to Reviewers’ comments:
Comment 1
Did the author test the markers for islet-like cells in the culture by immunostaining to visualize the islet-like cells? It would be better to visualize the islet-like cells in the groups to see whether the different conditions affect the differentiation of stem cells to islet-like cells.
Answer to comment 1
We performed immunostaining to islet-like cells in each culture condition such as Control, DC(2), DC(1), and DC(0), using insulin and glucagon. We observed the production of insulin and glucagon in islet-like cells in all culture conditions. In addition, we measured the area of the positive cells to calculate the percentage of insulin and glucagon produced cells. From the results, we found the percentage of insulin or glucagon positive cells of islet-like cells were significantly increased in DC(0) compared to Control culture condition, that consistent with the mRNA expression levels of INS and GCG. We now include this explanation, results on page 7, lines 304-325.
Comment 2
For the mRNA expression, what time point the authors collected samples? In Method 2.6, the author described “Cells were collected on day 6 in the stage 6 from each group after differentiation with TRIZol reagent”. This is confusing, it looks at day 6 of the stage 6? Or the author could use total days from starting.
Please include the time point for C-peptide.
Answer to comment 2
The samples for the RT-qPCR were collected on day of start (day 0) and last day (day 6) of the differentiation of endocrine progenitor cells into islet-like cells. We now corrected the confusing sentences (page 5).
Comment 3
Please include the time point for C-peptide.
Answer to comment 3
The time point of C-peptide assay was last day (day 6) of islet-like cell differentiation. We now corrected the confusing sentences (page 5).
Comment 4
The authors evaluated the short-term, how about the effect of long-term using this system?
Answer to comment 4
We have not yet tried dialysis suspension culture for more than 6 days, but since we did not observe any decrease in dialysis functions during the 6 days of culture in this work, we expect that dialysis suspension culture is possible for longer culture. However, there is a risk of depletion of growth factors, and accumulation of cytotoxic macromolecules such as inflammatory factors that cannot be removed by dialysis during long-term culture. Therefore, in long-term use of dialysis culture system, it might be necessary to increase the frequency of medium changes, and further studies are required for long-term use. We now include this explanation in the discussion briefly, on page 11, lines 432-437.
Reviewer 2 Report
The authors in this article address an important issue in the field of cell therapy, which is minimizing the cost of hPSC-derived islet-like cells. The authors also discuss that their improved method of utilizing a dialysis system to retain cytokines and growth factors in the differentiation media and removal of lactate from the media is beneficial for differentiating islet cells by increasing insulin expression and glucose-stimulated insulin secretion. However, multiple experiments are still required to support the authors’ claims in the article. Further experiments will also help the authors understand the mechanism by which beta cell functionality is improved.
- Flow cytometry should be performed for INS or C-PEPTIDE, GCG and other markers like NKX6.1 to confirm the increase in endocrine cells under different conditions like control, DC(0), DC(1) and DC(2), as well as to show if mono-hormonal insulin-expressing cells increase in DC(0) condition contributing to improved glucose-stimulated C-peptide secretion.
- Glucose-stimulated C-peptide secretion assay should also be represented as fold increase in C-peptide secretion under high glucose compared to low glucose for each condition. While the C-peptide secretion at 2 mM glucose is higher in DC(1) and DC(0) confirming the higher insulin expression in these conditions, the fold increase or stimulation index for DC(2) seems larger than DC(1) and more similar to DC(0) than to control. Also, authors should perform a second phase of glucose stimulation to assess C-PEPTIDE secretion.
- Functionality of insulin-secreting cells generated in control and DC(0)-(2) should be assessed by different methods such as measuring INSULIN secretion in response to other stimuli such as membrane depolarizing agents, voltage channel regulation and modulation of ADP/ATP levels etc., other than glucose.
- Ratio of C-PEPTIDE/PROINSULIN protein levels should be assessed for each condition to determine the extent of insulin biosynthesis and processing.
- Glucose levels in the dialysis medium seem to be unchanged from the data points on the graph in figure 2C, therefore the results mentioning that its decreased should be changed accordingly.
- Transcript levels for other islet genes could be assessed such as PDX1, NKX6.1, UCN3, GCK, GLUT1, PCSK1, PCSK2 amongst others.
Author Response
Answers to reviewer #2
Dear reviewer,
Thank you very much for appreciating our work and your valuable comments and suggestions. First, I would like to apologize for not being able to answer some of the comments properly. This manuscript was written based on an experiment operated several years ago, and due to the time and financial limitations, we were unable to prepare samples for proper analysis. We have attempted to answer your comments into consideration using alternative analysis as possible as we can. We sincerely apologize for this. Below we have replied to each comment in turn and marked the modified part. We look forward to your reply.
Best regards,
Hyunjin Choi
Point-to-point response to Reviewers’ comments:
Comment 1
Flow cytometry should be performed for INS or C-PEPTIDE, GCG and other markers like NKX6.1 to confirm the increase in endocrine cells under different conditions like control, DC(0), DC(1) and DC(2), as well as to show if mono-hormonal insulin-expressing cells increase in DC(0) condition contributing to improved glucose-stimulated C-peptide secretion.
Answer to comment 1
Since we do not have available samples for the flow cytometry, we performed immunostaining instead to measure the percentage of insulin and glucagon positive cells. The ratio of positive cell area to the cross-sectional area of islet-like cells was calculated using software IMAGE J. The results of the measurements showed a significant increase in the percentage of insulin positive cells in DC(1) and DC(0) relative to the Control, and a significant increase of glucagon in DC(0) relative to the Control. These results showed similar tendency to the mRNA expression levels of INS and GCG. We now include this explanation, page 7, lines 304-325.
Comment 2
Glucose-stimulated C-peptide secretion assay should also be represented as fold increase in C-peptide secretion under high glucose compared to low glucose for each condition. While the C-peptide secretion at 2 mM glucose is higher in DC(1) and DC(0) confirming the higher insulin expression in these conditions, the fold increase or stimulation index for DC(2) seems larger than DC(1) and more similar to DC(0) than to control. Also, authors should perform a second phase of glucose stimulation to assess C-PEPTIDE secretion.
Answer to comment 2
The reason for the lack of glucose responded insulin secretion in Control, DC(2), and DC(1) thought to be due to the low maturity of islet-like cells, particularly the maturity of glucose sensing and metabolism related function that is important for glucose simulated insulin secretion. Detailed explanation is described in page10, lines 396-437.
Like as you commented, glucose simulated C-peptide secretion of DC(2) looks more similar to DC(0) than control. Although we observed the tendency of upregulations in GLUT2 and the production rate of insulin in DC(2) compared to Control, we were not able to find any consistent tendencies in this study, might need further studies.
Actually, we performed 2 phase of glucose stimulation and the C-peptide secretion of phase 2 was measured. We thought the sentence was confusing, so we corrected sentence more clearly. Page 5, lines 199-235.
Comment 3
Functionality of insulin-secreting cells generated in control and DC(0)-(2) should be assessed by different methods such as measuring INSULIN secretion in response to other stimuli such as membrane depolarizing agents, voltage channel regulation and modulation of ADP/ATP levels etc., other than glucose.
Answer to comment 3
We were not able to perform the measurement, as we did not have appropriate samples and, we could not find other alternative method in this work. We sincerely apologize for this.
Comment 4
Ratio of C-PEPTIDE/PROINSULIN protein levels should be assessed for each condition to determine the extent of insulin biosynthesis and processing.
Answer to comment 4
We were not able to perform the measurement, as we did not have appropriate samples. However, from the mRNA expression level of PCSK1 related to the processing of proinsulin to C-peptide and insulin, there were no significant changes showed between all culture conditions. These results indicating that the insulin processing ability of islet-like cells in each culture conditions might be similar.
Comment 5
Glucose levels in the dialysis medium seem to be unchanged from the data points on the graph in figure 2C, therefore the results mentioning that its decreased should be changed accordingly.
Answer to comment 5
Thank you for your correction. We corrected sentence appropriately.
Comment 6
Transcript levels for other islet genes could be assessed such as PDX1, NKX6.1, UCN3, GCK, GLUT1, PCSK1, PCSK2 amongst others.
Answer to comment 6
We performed additional measurements of mRNA expression levels of PDX1, NKX6.1, GLUT1, GLUT2, GCK, and PCSK1. From the results, we could not find significant differences in the expression levels of β cells development related genes such as PDX1 NKX6.1, and hormonal processing related gene like PCSK1. On the other hand, we found the tendency of upregulations in glucose sensing and metabolism related genes, such as GLUT1, GLUT2 and GCK in DC(0). These results indicating that the improvement in the glucose stimulated insulin secretion ability of DC(0) is possibly due to the maturation of glucose sensing and metabolism function. We now include this explanation in the discussion, on page 11, lines 396-428.
Round 2
Reviewer 2 Report
The authors addressed some of the comments about investigating the characteristics of the generated beta cells by providing alternative assessments. Beta cell differentiation from hPSCs has improved greatly, with enhanced protocols and techniques being optimized. Generation of pancreatic beta cells require extensive evaluation to confirm that they are monohormonal and functional, most of which was lacking in the paper. The author’s protocol for differentiation only generated 9-25% of insulin-expressing cells, which is not enough to perform functional studies that could yield conclusive results. The gap and the necessity in the field is to improve differentiation methods to generate functional beta cells from hPSCs that can be used for cell therapy and disease modeling and it is still unclear whether the dialysis system for differentiation adds to this or not. The authors therefore must perform additional experiments in order to support their current results.
While authors touched on the limitation that while dialysis culture improves overall glucose-stimulated insulin, the frequency of medium change needs to be determined which requires further work in the Discussion; they should highlight this in the Abstract and Results as well.
The authors must look into employing more recent and robust beta cell differentiation protocols from hPSCs and performing those with their dialysis system in order to improve the total percentages of insulin expressing cells. It is likely that if they perform their analysis on a reasonable proportion of beta cells, their results for gene expression and GSIS will align more robustly.
- Authors should also provide explanation on why DC(0) and DC(1) have higher insulin and other metabolic gene expression but fold insulin secretion is not better than DC(2). DC(2), on the other hand, have less basal insulin secretion at 2mM glucose compared to DC(1) and DC(0) but the fold change under 20 mM is higher.
- GSIS fold change for DC(0) should not be similar to DC(2) but higher than that in order to support the authors’ claim that dialysis improves beta cell differentiation and to be consistent with their results, specifically, number of INS+ cells increased in DC(0) and DC(1) only.
- While presenting GSIS results, both first and second phase fold changes must be presented. Also, insulin secretion in response to KCl is crucial to determine the extent of membrane depolarization and total insulin content.
- Additionally, number of glucagon+ and insulin+ co-expressing beta cells should be determined along with NKX6.1+ and INSULIN+ co-expressing beta cells. This is crucial because it is unclear whether dialysis system improves the differentiation of the right kind of beta cells or the non-functional ones.
- GLUT-2 is not considered to be the major glucose transporter in human beta cells unlike rat beta cells. This should be corrected in the manuscript.
